# Sodium Butyrate Induces CRC Cell Ferroptosis via the CD44/SLC7A11 Pathway and Exhibits a Synergistic Therapeutic Effect with Erastin

**DOI:** 10.3390/cancers15020423

**Published:** 2023-01-09

**Authors:** Zhongbo Bian, Xiaodie Sun, Lulin Liu, Yong Qin, Qiuyu Zhang, Huahuan Liu, Lianzhi Mao, Suxia Sun

**Affiliations:** Department of Nutrition and Food Hygiene, Guangdong Provincial Key Laboratory of Tropical Disease Research, School of Public Health, Southern Medical University, Guangzhou 510515, China

**Keywords:** sodium butyrate, CRC, ferroptosis, CD44, SLC7A11

## Abstract

**Simple Summary:**

Sodium butyrate (NaB) is a short-chain fatty acid produced by intestinal microbial fermentation of dietary fiber. It has been shown to be effective in inhibiting colorectal cancer (CRC), but the mechanism is not known. We verified the ability of NaB to induce ferroptosis and the effect on relevant genotypes in normal intestinal cells and colorectal tumor cells, respectively. Moreover, better inhibition of tumor cells was observed when NaB was combined with Erastin (a ferroptosis-positive drug), suggesting that NaB combined with Erastin might have a stronger anti-CRC effect.

**Abstract:**

Colorectal cancer (CRC) is one of the most common malignancies, and effective treatment and prevention methods are lacking. Sodium butyrate (NaB) is a short-chain fatty acid produced by intestinal microbial fermentation of dietary fiber. It has been shown to be effective in inhibiting CRC, but the mechanism is not known. Methods: Human normal intestinal epithelial cell line FHT and colorectal tumor cell line HCT-116 were treated with NaB alone or in combination with different programmed cell death inhibitors. Cell activity was then assessed with MTT assays and PI staining; ferroptosis with Fe^2+^, glutathione (GSH), and lipid peroxidation assays; signaling pathway screening with PCR arrays; and CD44, SCL7A11, and GPX4 expression with Western blotting. A CD44-overexpressing HCT-116 cell line was constructed to determine the effect of the overexpression of CD44 on NaB-induced ferroptosis. The synergistic effect of co-treatment with NaB and Erastin was assessed by isobolographic analysis. Results: NaB induced apoptosis and ferroptosis in HCT-116 cells but only induced low-level apoptosis in FHC cells. Moreover, NaB significantly increased intracellular Fe^2+^ and promoted GSH depletion and lipid peroxidation in HCT-116 cells. Ferroptosis-related qPCR array analysis identified CD44/SLC7A11 as a potential effector molecular of NaB-induced ferroptosis. NaB significantly inhibited the expression of CD44 and SLC7A11 in mouse CRC tissues. A CD44 overexpressed HCT-116 cell line was used to verify that CD44/SLC7A11 was a key signaling pathway that NaB-induced GSH depletion, lipid peroxidation accumulation, and ferroptosis in HCT-116 cells. Examination of whether NaB can increase the effect of ferroptosis agents showed that NaB, in combination with Erastin, a ferroptosis inducer, further promoted HCT-116 cell death and increased changes of ferroptosis markers. Conclusions: Our results suggest that NaB induces ferroptosis in CRC cells through the CD44/SLC7A11 signaling pathway and has synergistic effects with Erastin. These results may provide new insights into CRC prevention and the combined use of NaB and ferroptosis-inducing agents.

## 1. Introduction

CRC is one of the most common malignancies worldwide. In 2020, there were about 1.88 million new cases of CRC and 915,000 deaths, making it the third-leading cause of morbidity and the second-leading cause of mortality [1]. The primary treatments for CRC are surgery, adjuvant radiotherapy, and chemotherapy. The comprehensive use of multiple treatments has improved the 5-year survival rate of patients with CRC, but the 5-year survival rate in low-income countries is <50% [2]. The high incidence of CRC is closely related to genetics, lifestyle, and poor diet [3]. The relation between dietary fiber intake and CRC is well-recognized: when the average dietary fiber intake is 35 g/d, the risk of CRC decreases by 40% [4,5].

Short-chain fatty acids (SCFA) are the products of microbial digestion of dietary fiber in the gut. They include acetic acid, propionic acid, and butyric acid, which are necessary for the metabolic needs of the colon and body in general. The antagonistic activity of butyric acid and butyrate is more clear than that of acetic acid and propionic acid. In tumor cells, butyrate mainly accumulates in the nucleus and acts as a histone deacetylase inhibitor (HDACi) to promote the expression of anticancer genes and inhibit the proliferation and apoptosis of tumor cells [6,7]. However, few studies have examined the role of sodium butyrate (NaB) in the regulation of ferroptosis.

Ferroptosis is a newly recognized type of programmed cell death dependent on iron and characterized by lipid hydroperoxide (LPO) accumulation; it is different from apoptosis, necrosis, and autophagy [8]. In mammalian cells, the membrane is the main site affected by lipid peroxidation during ferroptosis due to the abundant presence of phospholipids containing polyunsaturated fatty acids. The cystine/glutamate antiporter system (System Xc-) and glutathione peroxidase 4 (GPX4) are considered key molecules in the regulation of lipid peroxidation and ferroptosis [9,10,11]. System Xc- consists of a light chain subunit, SLC7A11, and a heavy chain subunit, SLC3A2, and mediates the exchange of extracellular cysteine and intracellular glutamate, thereby promoting the synthesis of intracellular glutathione (GSH) which protects cells from damage due to oxidative stress. In addition, studies have shown that CD44 can stabilize the SLC7A11 subunit on the cell membrane, promote the growth of CRC cells, and increase SLC7A11-dependent protection from reactive oxygen species (ROS), thus enabling cancer cells to escape from oxidative damage [12,13]. Thus, the targeted inhibition of CD44/SLC7A11 may make tumor cells prone to ferroptosis.

Our previous studies showed that NaB could induce autophagy and apoptosis in CRC cells. However, there has been no research regarding NaB inducing ferroptosis and thus inhibiting CRC. Therefore, the purpose of this study was to investigate if NaB induces ferroptosis in CRC cells and the mechanism by which it occurs.

## 2. Materials and Methods

### 2.1. Reagents and Chemicals

NaB was purchased from Sigma-Aldrich (St. Louis, MI, USA); molecular formula, C4H7O2Na; molecular weight 110.09; high-performance liquid chromatography analysis of standards ≥ 98%. Dulbecco’s modified Eagle’s medium (DMEM), fetal bovine serum (FBS) and 0.25% Trypsin solution with EDTA were purchased from Gibco (USA). Erastin, Necrostatin 1 (Nec-1) and Z-VAD-FMK inhibitors were purchased from Slleck (Shanghai, China). Ferrostatin-1(FER-1), chloroquine (CQ), acetylcysteine (NAC), 3-(4,5-dimethyl-2-thiazolyl)-2,5-diphenyl-2-H-tetrazolium bromide (MTT), and dimethyl sulfoxide (DMSO) were purchased from Sigma-Aldrich (USA).

### 2.2. Cell Culture

Normal human colon epithelial cell FHC and human colorectal tumor cell HCT-116 were purchased from the Shanghai Biological Cell Research Library of the Chinese Academy of Sciences. HCT-116 and FHC cells were cultured with DMEM supplemented with 10% FBS, penicillin (100 U/mL), and streptomycin (100 μg/mL) in a moist environment of 37 °C and 5% CO_2_.

### 2.3. MTT

The viability of HCT 116 and FHC cells was measured using the MTT method. Cells are placed in 96-well plates (8 × 10^3^ cells/well) overnight and then treated with the corresponding drug for 24–72 h. Add 20 μL of MTT (5 mg/mL) per well and incubate for another 4 h. DMSO is then added to dissolve the MTT tetrazolium crystals. Place the plate on an enzyme immunoassay detector and shake for 10 min under low-speed shaking. Absorbance is obtained at an optical density (OD) of 490 nm.

### 2.4. Cell Morphology Observation and Photography

Cells were seeded in 6-well plates and treated with the corresponding drug for 24–72 h. After removing the 6-well plate, discard the old full medium, wash with PBS twice, carefully drain the remaining liquid with a pipette, fix the cells with methanol for about 15–20 min at room temperature and then dump them, and carefully wash with PBS twice. Observe the morphology and number of cells under the white light lens of the microscope and take pictures.

### 2.5. PI Staining and Fluorescence Mapping

Collect adherent and suspension cells during the experiment, centrifuge at 1000 rpm, 3 min, wash twice using 1 mL PBS, count cells under the microscope, adjust the number of cells to about 1 million cells, and resuspend cells with 500 μL of stain solution after centrifugation again. After 5 min of incubating in the light, add 1 drop of the cell suspension to the slide, cover the coverslip and observe its red fluorescence and take a picture.

### 2.6. Ferroptosis PCR Array Analysis and Real-Time Fluorescence Quantitative PCR

The PCR array used in the study was customized to the Shanghai WCGENE gene by the research group according to the research needs. The size of the PCR array was 96 Wells, and the gene sequence of each well was verified by WCGENE. The list of genes was shown in Table 1.

Total RNA was isolated and purified using TRIzol, phenol, chloroform and ethanol. After RNA reverse transcription, mix 900 μL of cDNA and SYBR^®^ Premix Ex TaqTMI according to the instructions, mix well, add the mixture (9 μL per well) into the 96-well plate, seal the plate with transparent sealing plate membrane after adding, and centrifuge again. A real-time PCR detection system (CFX Connect™, BIORAD, Hercules, CA, USA) was used to detect gene amplification. The method was also used for real-time fluorescence quantitative PCR, and the primer sequences were shown in Table 2.

After calculating the expression results of each gene, they need to be sorted out into tables (Appendix A) and sent into Rstudio for heat map and volcano map drawing. Geometric herewith’s figure, GO pathway enrichment and enrichment of and function network with the protein interaction network analysis respectively by Bioinformatics at http://www.bioinformatics.com.cn (accessed on 14 February 2022) and Metascape at https://metascape.org/gp/index.html (accessed on 14 February 2022), there are two free online platforms for data analysis and visualization.

### 2.7. Population Public Database Analysis

The box plot, violin plot, and scatter plot of CD44 and SLC7A11 gene expression in this part of the results were drawn by the TCGA and GTEx visualization website GEPIA2 http://gepia2.cancer-pku.cn/#index (accessed on 14 February 2022).

The population viability analysis was drawn by the generic cancer prognosis PROGgeneV2 http://www.progtools.net/gene/index.php database (accessed on 14 February 2022).

### 2.8. Western Blotting

HCT 116 and FHC cells were lysed with RIPA solution (Beyotime Biotechnology, Shanghai, China) containing protease inhibitors (Keygen, Nanjing, China) after treatment with appropriate drugs. Total protein was collected by centrifugation at 14,000× *g* for 15 min. The protein samples were separated by 10% sodium dodecyl sulfate (SDS) -polyacrylamide gel electrophoresis (PAGE) at 80 V and then transferred to a polyvinylidene fluoride (PVDF) membrane (Bio-RAD Laboratories). The membrane was then incubated with primary and secondary antibodies to detect target proteins. In the final step, the membrane was visualized using a chemiluminescence detection kit (Pierce Biotechnology, Rockford, IL, USA) according to the manufacturer’s instructions. The target protein was standardized as glyceraldehyde -3-phosphate dehydrogenase (GAPDH) and used as a sample control.

### 2.9. Immunofluorescence and Laser Confocal Imaging

The cells in the confocal petri dish were fixed, permeated and sealed in advance. According to the antibody instructions, the corresponding primary antibody and fluorescent secondary antibody and DAPI dye were used for detection (SLC7A11/xCT rabbit Polyclonal antibody dilution ratio was 1:5000, CD44 mouse monoclonal antibody dilution ratio was 1:4000, Alexa Fluor 488 rabbit antibody or Alexa Fluor 594 mouse antibody dilution ratio of 1:1000). Photographs were taken using a confocal microscope.

### 2.10. Animal Tissue Immunohistochemistry and Prussian Staining

Animal experimental design and tissue sampling are described in our previously published study [14]. The subsequent immunohistochemical and Prussian staining experiments were commissioned by BIOS Biological (Guangzhou, China). CD44 Rabbit mAb (ProteinTech, Wuhan, China) and SLC7A11/xCT Rabbit mAb (ProteinTech, China) diluted at 1:50 are used. Then, 3 non-overlapping fields were randomly selected for each slice under a 40× microscope for macroscopic observation comparison.

### 2.11. GSH Detection

Cells were lysed with Glutathione Buffer, and the supernatant was collected. The GSH content in the supernatant was assessed by a reduced glutathione assay kit (Solarbio, Beijing, China, #BC1175) according to the manufacturer’s instructions. The absorbance was measured at 450 nm (A450). The concentration of GSH was determined using a standard GSH calibration curve and was related to the number of cells used.

### 2.12. Fe^2+^ Staining

Serum-free basal medium DMEM and FerroOrange working solution were added to the petri dish so that the final concentration of FerroOrange in the basal medium was 1 μM. The cells were then incubated in an incubator for 30 min in darkness. After incubation, the six-well plate was taken out, and the red fluorescence intensity was observed directly with a fluorescence microscope. FerroOrange: Ex: 561 nm, Em: 570–620 nm.

### 2.13. Lipid ROS Detection

The cells were washed twice with PBS and then added with serum-free DMEM basal culture medium and C11 BODIPY 581/591 dye so that the final concentration of C11 BODIPY 581/591 was 10 μM. The cells were incubated in a cell culture box for 30 min without light. After incubation, the cells were taken out and washed with PBS more than 2 times. A confocal fluorescence microscope was used to stimulate the cells with 488 nm and 565 nm lasers, respectively, and relevant fluorescence signal images were collected.

### 2.14. Isobolographic Analysis

After drug combination treatment and MTT assay, isobolographic analysis was performed with reference to the method of Huang et al. [15]. First, a cartesian coordinate system was set up, in which the X-axis and Y-axis represent the doses of the two drugs (Erastin and NaB in this study) to achieve the same efficacy. In addition, A and B, respectively, represent the dose of IC50 when the two drugs are used alone. The two points and the origin can be enclosed into an equivalent triangle. If the two drugs act synergistically, the equivalent points of drug combination fall within the range of an equivalent triangle.

### 2.15. Statistical Methods

Statistical analyses of different sets of experiments were performed using Prism 7 (GraphPad Software, Inc., La Jolla, CA, USA). A 2-tailed *t*-test or Mann–Whitney’s test was used in a single variable with a 2-group comparison, a 1-way ANOVA with Tukey’s post-test or a Kruskal–Wallis test was used in single-variable comparisons with more than 2 groups, and 2-way ANOVA with Bonferroni posttest for multivariable analyses. Differences with *p* <  0.05 were regarded as statistically significant.

## 3. Results

### 3.1. NaB Induces CRC Cell Death through Apoptosis and Ferroptosis

To test our hypothesis that ferroptosis may occur during NAB-induced CRC cell death, we used z-VAD (an apoptosis inhibitor), ferrostatin-1 (Fer-1, a potent ferroptosis specific inhibitor), necrostatin-1 (Nec-1, a potent necrotizing apoptosis inhibitor), and chloroquine (CQ, an effective autophagy inhibitor) to determine the types of programmed cell death induced by NaB. First, we found that NaB promotes FHC cell proliferation at low concentrations but inhibits it at high concentrations. In contrast, NaB had a more obvious inhibitory effect on HCT-116 cells. There were also differences in the response of FHC cells and HCT-116 cells to different cell death inhibitors. In FHC cells, only the apoptosis inhibitor z-VAD could partially reverse cell viability inhibited by NaB. In HCT-116 cells, the apoptosis inhibitor z-VAD and ferroptosis inhibitor Fer-1 partially reversed the proliferation of HCT-116 cells inhibited by NaB, while autophagy inhibitor CQ further inhibited the proliferation of HCT-116 cells. Nec-1 did not affect either cell line (Figure 1A–D). Next, we observed cell morphology and death of FHC cells and HCT-116 cells when different types of cell death inhibitors were combined with NaB, and the results were consistent with the previous results (Figure 1E).

To further confirm the role of NaB in ferroptosis in CRC cells, levels of GSH, lipid ROS, and Fe^2+^ were measured. The results showed that 10 mM NaB significantly down-regulated the GSH content in HCT-116 cells but did not affect the GSH level in FHC cells. The addition of N-acetyl-L-cysteine (NAC), a precursor of GSH, and Fer-1 significantly reversed the reduction of GSH induced by NaB in HCT-116 cells, but the apoptosis inhibitor z-VAD had no protective effect on the depletion of GSH induced by NaB (Figure 1F,G). A C11-BODIPY fluorescence probe was used to detect lipid ROS, and the findings were consistent with the results of intracellular GSH detection (Figure 1H). The FerroOrange ferrous ion fluorescence probe was used to measure the Fe^2+^ content in FHC cells and HCT-116 cells to determine if ferroptosis was occurring. HCT-116 cells had a higher basal Fe^2+^ level than FHC cells, and the intracellular Fe^2+^ level increased after treatment with NaB for 24 h; no significant change in the level of Fe^2+^ in FHC cells was observed (Figure 1I). 

Based on these results, we hypothesized that NaB caused CRC cell death by inducing apoptosis and ferroptosis in CRC cells but did not induce ferroptosis in normal intestinal epithelial cells.

### 3.2. Ferroptosis-Related qPCR Array Analysis Identified CD44/SLC7A11 as a Potential Effector Molecular of NaB-Induced Ferroptosis

Database FerrDb (http://www.zhounan.org/ferrdb/legacy/index.html, accessed on 14 February 2022) is a collection of authority management ferroptosis-related markers, regulatory factors, and associated diseases. We selected 92 genes for ferroptosis-related gene array analysis based on reliability scores. According to the analysis, there were 18 differentially expressed genes in FHC, and HCT-116 cells and 10 differentially expressed genes in HCT-116 cells after treatment with 10 mM NaB for 24 h (*P*adj < 0.05, Log2 fold-change [FC] > 1). The genes are illustrated in a gene heat map and volcano map (Figure 2A,B). All the differentially expressed genes (*p* < 0.05) in the HCT-116 cell control group and HCT-116 cell NaB treatment group were used to perform functional pathway annotation of the Gene Ontologies (GO) database. GO analysis results include molecular function (MF), cellular component (CC) and biological process (BP). The enrichment score of BP was the highest among the 3 analyses, and the annotation results showed that NaB significantly regulated functions related to GSH and Fe^2+^ metabolism (Figure 2C). Therefore, in the PCR array, we sorted out all genes related to Fe^2+^ and GSH metabolism. The results showed that NaB changed the expression of GSH-related genes *GCLC*, *GLS2*, *GOT1*, *CHAC1*, *BECN1*, *SLC7A11*, *CD44* and iron-related genes *IREB1*, *IREB2*, *TFR1*, *SLC40A1*, *NCOA4*, and *ATG5* in HCT-116 cells (Figure 2D). 

In order to further study the relations between all pathways and functions, all the differentially expressed genes of HCT-116 cells in the control group and HCT-116 cells in the NaB treatment group were presented as a network diagram, in which items with similarity > 0.3 were connected by edges (Figure 2E). Protein-protein interaction enrichment analysis was performed to obtain the total protein network. Molecular Complex Detection (MCODE) identified HRAS, NRAS, KRAS, SLC7A11, and CD44 and VDAC3, ALDH1A1, GOT1, and CS as 2 groups of key central protein interaction networks (Figure 2F). 

Previous studies have found that CD44 and SLC7A11 are key molecules that affect the transport of extracellular cystine for the intracellular synthesis of GSH and regulate ferroptosis and that they may be interrelated. Therefore, we determined if NaB induced ferroptosis in CRC cells via CD44/SLC7A11.

### 3.3. CD44 and SLC7A11 Are Highly Expressed and Positively Correlated in Human CRC

In order to confirm the accuracy of the PCR array analysis and explore the role and correlation of CD44 and SLC7A11 in human CRC, population gene expression and survival analysis were performed using the pan-cancer data analysis tools GEPIA and ProGgene V2. The results showed that compared with a healthy population, *CD44* and *SLC7A11* mRNA levels were higher in CRC patients, but there was no difference in the expression levels of *CD44* and *SLC7A11* in different stages of CRC (Stage I to Stage IV) (Figure 3A). The general linear correlation model showed that *CD44* was positively correlated with *SLC7A11* expression in CRC patients (Figure 3B) (Pearson correlation coefficient = 0.21, *p* < 0.001). Survival analysis results showed that CRC patients with high expression levels of CD44 and SLC7A11 had a worse prognosis and shorter survival time (Figure 3C).

### 3.4. NaB Inhibits CD44/SLC7A11 Expression In Vivo and In Vitro

Targets of ferroptosis induced by NaB were studied in vivo and in vitro. Western blotting results showed that CD44 and SLC7A11 exhibited low expression in FHC cells and high expression in HCT-116 cells (Figure 4A). NaB significantly inhibited CD44 and SLC7A11 protein expression in HCT-116 cells but did not affect the expression of GPX4 protein. NaB did not affect the expression of CD44, SLC7A11, and GPX4 in FHC cells (Figure 4B). Confocal laser study results showed that after treatment with 10 mM NaB for 24 h, the expressions of CD44 and SLC7A11 in HCT-116 cells were significantly down-regulated, and co-localization was inhibited (Figure 4C). These results indicate that NaB inhibits the expression of key ferroptosis proteins CD44 and SLC7A11 in HCT-116 cells but does not affect GPX4.

To verify the role of NaB in vivo, we established an AOM/DSS-induced murine inflammatory CRC model. 0.1M NaB(oral administration) reduces tumor load and tumor size in AOM/DSS-induced mice. Corresponding results and experimental methods are shown in our previous study [14]. Colorectal sections of the mice (a tumor-prone site) were examined by immunohistochemistry (IHC) staining and iron-specific staining (Prussian blue) for the key proteins CD44 and SLC7A11. IHC staining results showed CD44 and SLC7A11 expression was significantly up-regulated in the CRC model group, and the expression of CD44 and SLC7A11 was significantly decreased after treatment with NaB (Figure 4D). Prussian blue staining showed that treatment with NaB increased iron deposition at the tumor site (Figure 4D). These results indicate that NaB can effectively inhibit the expression of CD44 and SLC7A11 in a murine CRC model, which is consistent with the results of the cell-line experiments.

### 3.5. CD44 Overexpression Inhibits NaB-Induced Ferroptosis in CRC Cells

To further validate the role of CD44 and SLC7A11 in NaB-induced CRC ferroptosis, we transfected HCT-116 cells with a pcDNA3.1 (+) carrier construction of CD44 expression (CD44OE) or blank (vector) plasmid. The results showed that CD44 and SLC7A11 mRNA and protein levels in the CD44 overexpression group were significantly higher than those in the control and vector transfection groups (Figure 5A,B). Moreover, NaB reversed the mRNA and protein inhibition of CD44 and SLC7A11. Next, cell viability, GSH content change, and lipid ROS accumulation were studied to determine the occurrence of ferroptosis. The results showed that CD44 overexpression significantly restored the proliferation activity and cell morphology of cells treated with NaB for 24 h (Figure 5C–E). Although CD44 overexpressed plasmid transfection did not significantly increase the basal level of GSH in HCT-116 cells, the decrease in intracellular GSH content caused by NaB treatment was significantly alleviated in the CD44 overexpression group (Figure 5F). Similarly, after treatment with 10 mM NaB, lipid ROS accumulation was significantly increased in the control and vector-transfected cells, while significantly reduced lipid ROS accumulation was observed in CD44-transfected cells (Figure 5G). These results confirmed that the CD44/SLC7A11 signaling pathway is a key regulatory pathway of NAB-induced ferroptosis in HCT-116 cells.

### 3.6. Erastin and NaB Synergistically Induce CRC Cell Ferroptosis 

Several studies have shown that a histone deacetylase inhibitor (HDACi) combined with ferroptosis-inducing agents can enhance its antitumor effect [16,17,18]. Our results suggest that a high concentration or long-term exposure to NaB may promote apoptosis in normal intestinal epithelial cells. As such, it is important to optimize the concentration and exposure time of NaB to increase its antitumor effect and reduce the effects on normal colon cells. Erastin is a classic ferroptosis-inducing agent, and a concentration of about 10 μM induces ferroptosis in a number of tumor cell lines [8]. Results of our MTT assays showed that 10 μM Erastin did not significantly affect cell viability, but cell viability decreased rapidly with increasing NaB concentration in the Erastin-NAB treatment group (Figure 6A). Cell morphology examination and staining of dead cells also indicated enhanced cytotoxicity by Erastin-NAB combined treatment (Figure 6B). 

The effect of Erastin combined with NaB was further tested by isobolographic analysis (Figure 6C). MTT assay results showed that the IC50 and 95% confidence interval (CI) of NaB and Erastin for inhibiting HCT-116 proliferation were 12.420 mM (95% CI: 11.936–12.941 mM) and 26.978 µm (95% CI: 25.714–28.381 µm), respectively. The combination of NaB and Erastin at a concentration ratio of 12.420 mM to 26.978 µm exhibited a synergistic effect on the inhibition of proliferation of HCT-116 cells (IC50 = NaB: 2.43 mM + Erastin: 4.861 µm). Combined treatment with NaB and Erastin did not affect the proliferation activity of FHC cells(Figure 6D). In addition, compared with cells treated with NaB alone, the combined treatment group exhibited decreased GSH levels, increased accumulation of lipid peroxidation products, and significantly decreased levels of ferroptosis marker protein SLC7A11 in HCT-116 cells (Figure 6E–G). These results suggested that NaB and Erastin can synergistically induce ferroptosis in HCT-116 cells.

## 4. Discussion

Butyrate produced by dietary fiber promotes colon health and has anti-tumor properties. Butyrate is the preferred energy source in normal colon cells and can maintain mucosal integrity and inhibit inflammation and the development of cancer by affecting immunity, balancing intestinal flora, and regulating gene expression [7,19]. In tumor cells, butyrate acts as HDACi to promote the expression of anticancer genes, inhibit the proliferation of tumor cells, and promote apoptosis [20]. Recent pharmacotranscriptome and molecular biology experimental results suggest that HDACi drugs have the potential to regulate ferroptosis [18]. In a review of the synergistic effects of diet and gut microbes to enhance programmed death in CRC, Chapkin et al. suggested that butyrate and omega-3 polyunsaturated fatty acids may be effective in inducing ferroptosis. This is believed to be due to ROS accumulation induced by butyrate and the entry of long-chain polyunsaturated fatty acids into cell membranes [21]. However, no studies have confirmed the ability of NaB to induce ferroptosis in normal intestinal epithelial or CRC cells at low and high concentrations. In this study, we found that a high concentration of NaB can induce a high degree of apoptosis in CRC cells and a low degree of apoptosis in normal colon cells, and NaB only induces ferroptosis in tumor cells.

Iron is an active redox metal that is involved in free radical formation and expansion of lipid peroxidation. Generally, cancer cells require excess iron to support their high metabolic rate and support this phenotype by altering cellular pathways. Therefore, an increased iron level can make tumor cells more prone to ferroptosis [22,23]. The PCR array results in this showed that NaB significantly upregulated the iron-related genes TFR1 and IREB1/2. IREB1/2 is a primary binding protein of cells when iron is needed; when cells require iron, IREB1/2 directly binds to the RNA stem-loop structure in the 3′-untranslated region (UTR) of TFR1 mRNA and stabilizes its expression, thereby increasing intracellular iron concentration [24]. In addition, the up-regulation of ferritinophagy-related genes NCOA4 and ATG5 also suggests that NaB may degrade ferritin and release iron ions through autophagy [25].

Our results showed that both the GSH synthesis rate-limiting enzyme GCLC and the GSH cleavage-related gene CHAC1 were significantly up-regulated by NaB, suggesting that NaB plays a role in the regulation of GSH synthesis and metabolism [26,27]. This is in addition to the role of the key enzymes GOT1 and GLS2 in the metabolism and utilization of glutamine, an important precursor of GSH. More importantly, SLC7A11, a key protein in the formation of System Xc- which transports cysteine into the cell to synthesize GSH, was significantly inhibited by NaB at both the mRNA and protein levels. This suggests that NaB may effectively inhibit intracellular GSH content through this pathway. 

Studies have shown that CD44 promotes the deubiquitination of SLC7A11 by promoting the binding of the deubiquitination enzyme OTUB1 to SLC7A11, thus making SLC7A11 on the cell membrane stable against proteasome pathway degradation [28]. Our results suggest that NaB regulates CD44/SLC7A11-mediated GSH depletion and ferroptosis. GPX4 is another key regulator of ferroptosis and needs to acquire electrons from GSH for its antioxidant capacity. The PCR and protein assay results in this study showed that NaB does not have a role in GPX4 regulation.

Since colorectal tumors are often insensitive to traditional ferroptosis inducers, finding effective drug synergies has become an area of active research in the development of ferroptosis cancer treatments. For example, dihydroartemisinin (DAT) combined with RSL3, the HDAC inhibitor vorinostat combined with Erastin, and the DHODH inhibitor brequinar combined with sulfapyridine all induce ferroptosis in tumor cells [29,30,31]. Yang et al. reported that high expression of SLC7A11 may be related to the sensitivity of CRC cells to ferroptosis [18]. In our preliminary study, Erastin did not effectively inhibit SLC7A11 mRNA and protein levels in HCT-116 cells (data not presented), even though its pharmacological mechanism is explained by its effect on mitochondrial VDAC and System Xc- activity on the cell membrane. We found that the IC50 of Erastin on H-116 cells was over 20 µM for 24 h, which is much higher than that of other tumor cell lines (1–10 µM). However, NaB effectively inhibits SLC7A11 transcription and protein expression in HCT-116 cells. When HCR-116 cells were co-treated with NaB and Erastin, SLC7A11 protein levels were significantly down-regulated as compared to when cells were treated with NaB or Erastin alone. These findings may provide a basis for the development of combination drugs based on an HDACi and ferroptosis inducer.

## 5. Conclusions

The results of this study showed that NaB induces apoptosis and ferroptosis in CRC cells. Treatment with NaB significantly changed the expression of iron and GSH metabolite function-related genes. The findings also suggest that NaB may regulate ferroptosis in CRC cells through the CD44/SLC7A11 pathway. Finally, when NaB was combined with Erastin, inhibition of CRC cells was enhanced, suggesting that NaB combined with Erastin might have a stronger anti-CRC effect.

## Figures and Tables

**Figure 1 cancers-15-00423-f001:**
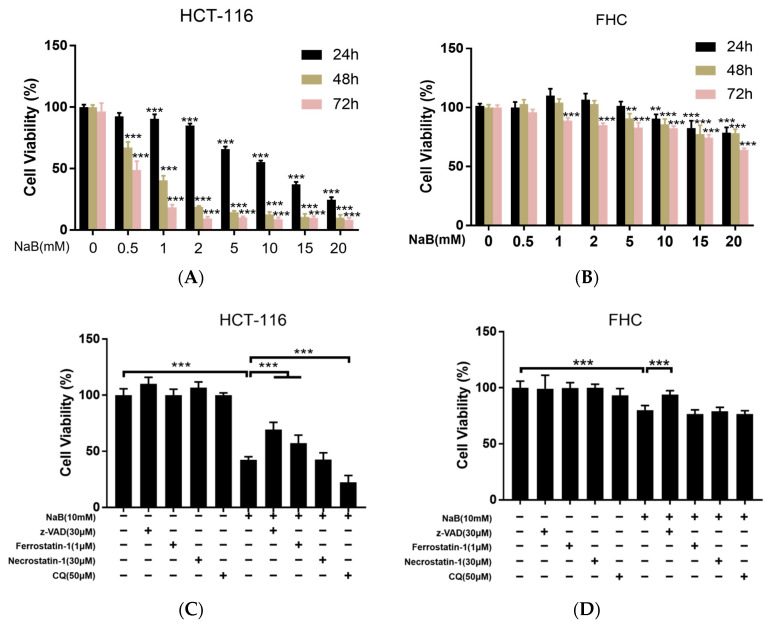
NaB induces colorectal cancer cell death via apoptosis and ferroptosis. (**A**,**B**) Growth curves of FHC and HCT-116 cells were examined using the MTT assay. Cell Cells were cultured in 0, 0.5, 1, 2, 5, 10, 15, and 20 mM NaB for 24, 48 and 72 h, and then viability was determined. (**C**,**D**) Effect of z-VAD (30 µM), ferrostatin-1 (1 µM), necrostatin-1 (30 µM), and CQ (50 µM) on cell viability when cultured with NaB (10 mM). All drug treatments were for 24 h. (**E**) Microscopy showing cell morphology and cell death. Upper panel: phase-contrast; lower panel, propidium iodide (PI) staining. (**F**,**G**) Quantification of reduced cellular GSH levels using the Micro Reduced GSH Assay Kit of cells cultured with NaB (10 mM) with and without z-VAD (30 µM), ferrostatin-1 (1 µM), and NAC (5 mM). (**H**) Lipid peroxidation was assessed in FHC cells and HCT-116 cells after exposure to NaB with and without z-VAD or ferrostatin-1 by a fluorescence microscope using C11- BODIPY. (**I**) Fe^2+^ accumulation was assessed in FHC cells and HCT-116 cells after exposure to NaB by fluorescence microscope using FerroOrange. The bar graph shows the mean ± standard deviation of 3 independent experiments. ** *p* < 0.01, *** *p* < 0.001.

**Figure 2 cancers-15-00423-f002:**
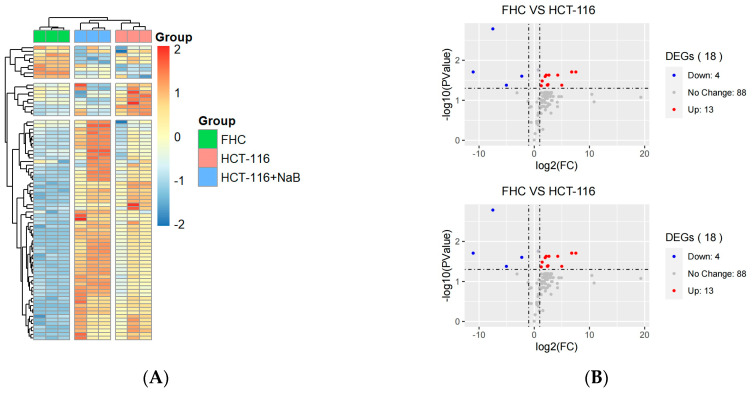
Ferroptosis-related qPCR array identified CD44 and SLC7A11 as potential effector molecules of NaB-induced ferroptosis (**A**) Expression trend of 88 ferroptosis-related genes in FHC cells, HCT-116 cells, and NaB-treated HCT-116 cells. (**B**) Volcano plots were used to identify differentially expressed genes between FHC cells vs. HCT-116 cells; HCT-116 cells vs. NaB-treated HCT-116 cells. (**C**) Gene ontology results of 3 ontologies. (**D**) BP cnetplot. (**E**) GSH and iron-related genes. (**F**) Analysis of the process enrichment network and protein interaction network. * *p* < 0.05, ** *p* < 0.01,*** *p* < 0.001.

**Figure 3 cancers-15-00423-f003:**
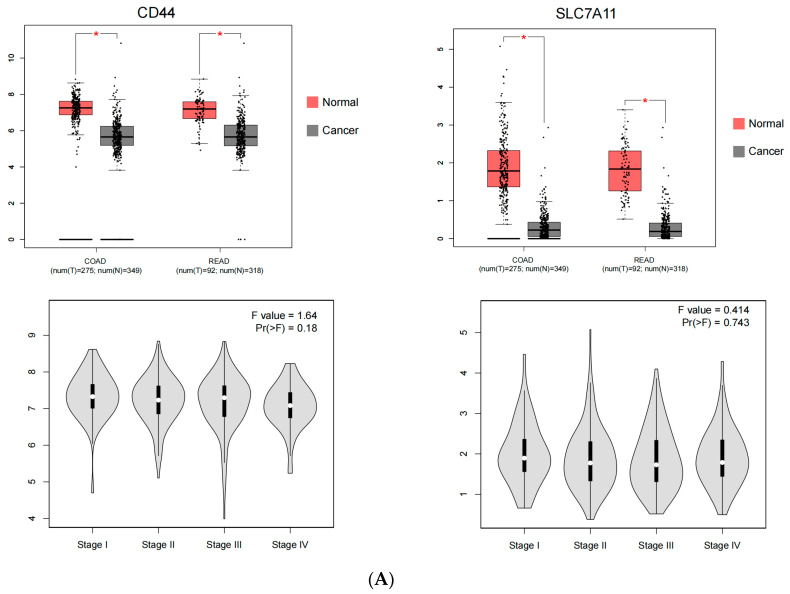
CD44 and SLC7A11 are highly expressed and positively correlated in human colorectal cancer. (**A**) Box diagram and stage violin diagram. (**B**) Scatter diagram. (**C**) Survivorship curve. * *p* < 0.05.

**Figure 4 cancers-15-00423-f004:**
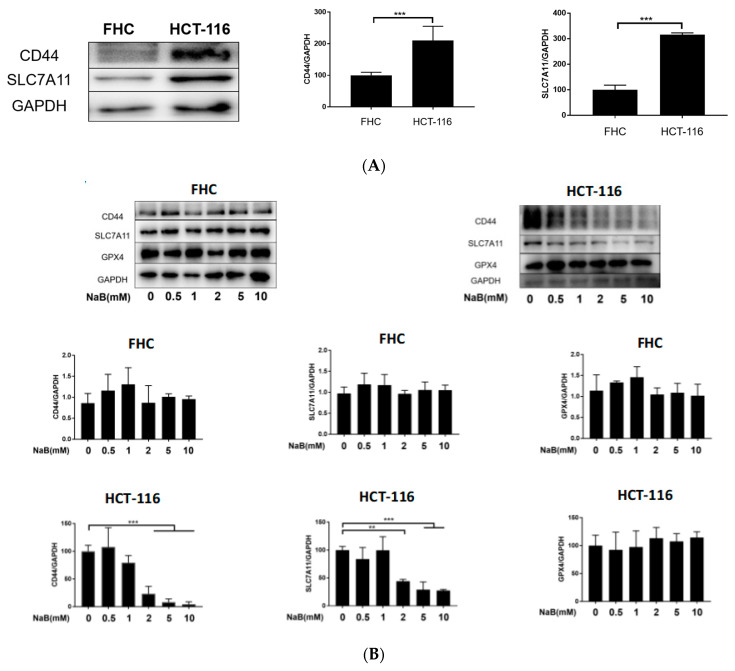
NaB inhibits CD44/SLC7A11 expression in vivo and in vitro. (**A**,**B**) Western blotting and quantitative analysis (The original Western blotting figures are in Appendix A). (**C**) Laser confocal microscopy and quantitative analysis. (scale bar = 50 µM). (**D**) Protein immunohistochemistry and Prussian blue staining (scale bar = 100 µM). * *p* < 0.05, ** *p* < 0.01, *** *p* < 0.001.

**Figure 5 cancers-15-00423-f005:**
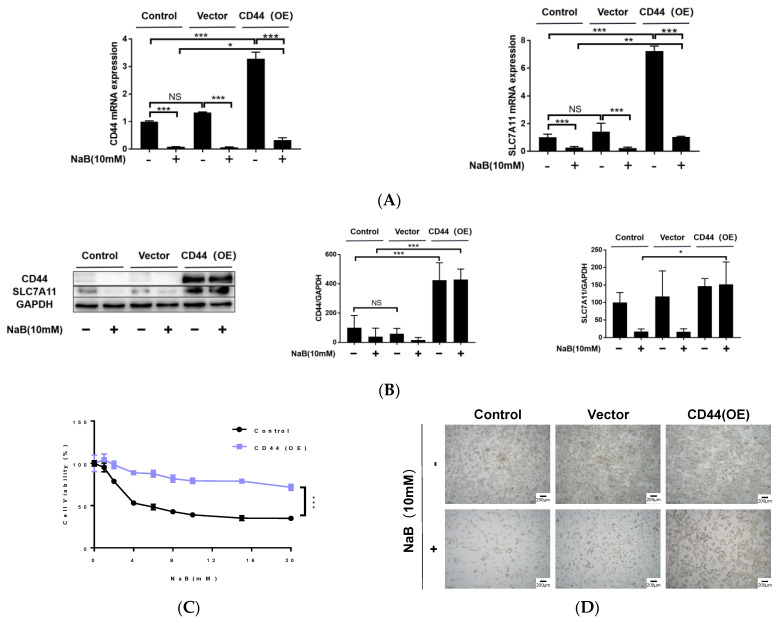
CD44 overexpression inhibits NaB-induced ferroptosis in colorectal cancer cells (**A**,**B**) CD44 and SLC7A11 mRNA and protein levels measured by RT-PCR and Western blotting (The original Western blotting figures are in Appendix A). GAPDH was used as the loading control. (**C**) Growth curves of HCT-116 cells and HCT-116 CD44 overexpression (OE) cells by MTT assay. Cell growth was inhibited by NaB and partially reversed by CD44 overexpression. (**D**,**E**) Microscopy showing cell morphology and cell death. (**D**) phase-contrast; (**E**) (PI) staining. (**F**) intracellular GSH content. (**G**) C11 detection of intracellular lipid peroxides. * *p* < 0.05, ** *p* < 0.01, *** *p* < 0.001.

**Figure 6 cancers-15-00423-f006:**
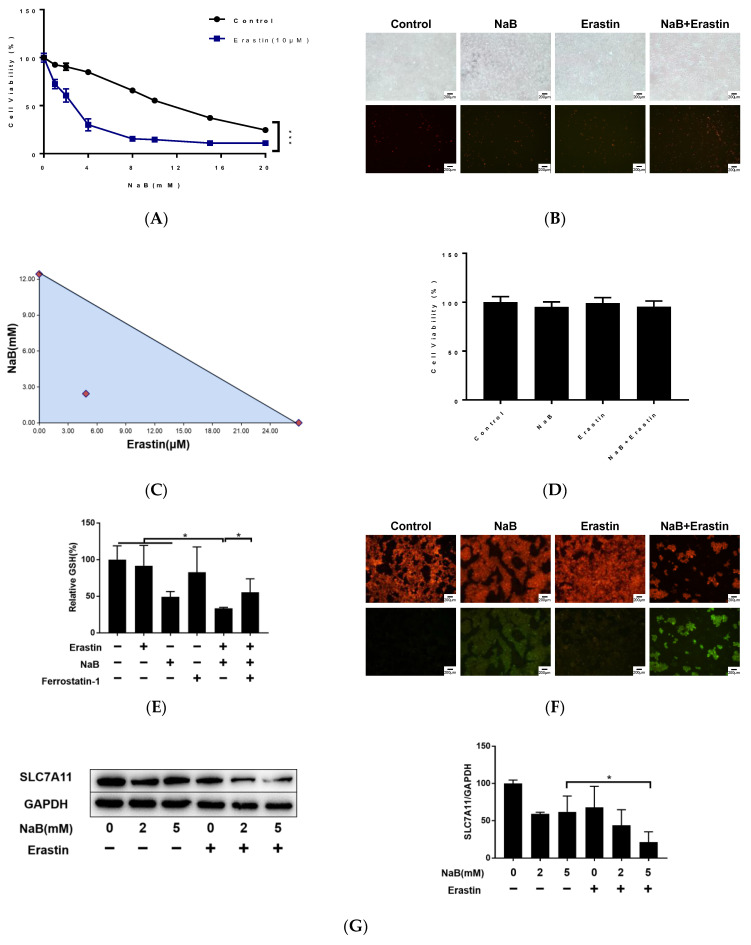
Erastin and NaB synergistically induce colorectal cancer cell ferroptosis. (**A**,**D**) MTT assay to detect cell viability. (**B**) Microscopy showing cell morphology and cell death. Upper panel: phase-contrast; lower panel, propidium iodide (PI) staining (scale bar = 100 µM). (**C**) Isobolographic analysis of drug combination. (**E**) intracellular GSH content. (**F**) C11 detection of intracellular lipid peroxides. (**G**) Western blotting and quantitative analysis (The original Western blotting figures are in Appendix A). * *p* < 0.05, *** *p* < 0.001.

**Table 1 cancers-15-00423-t001:** Total gene list.

**HUMAM**	**1**	**2**	**3**	**4**	**5**	**6**
A	IREB1	ATG5	CDO1	DPP4	GCLM	HARS
B	ACSL4	ATP5G3	CHAC1	ELAVL1	GLS2	HEPH
C	AKR1B1	BBC3	CISD1	EMC2	GOT1	HFE
D	AKR1B10	BECN1	CISD2	EPRS	GPX4	HMOX1
E	AKR1C1	BRAF	CP	FTH1	GSS	HMOX2
F	ALDH1A1	BRD4	CS	FTL	GSTA1	HRAS
G	ALOX15	CA9	CYBB	FTMT	GSTP1	HSF1
H	ATF4	CARS1	DMT1	GCLC	HAMP	HSPB1
**HUMAM**	**7**	**8**	**9**	**10**	**11**	**12**
A	IREB2	NCOA4	PCBP1	SAT2	STEAP3	VDAC2
B	KEAP1	NFE2L2	PCBP2	SLC1A5	STIM1	VDAC3
C	KRAS	NOX1	PPARG	SLC39A14	TF	MAP1LC3C
D	LOX	NOX3	PRDX6	SLC39A8	TFR1	PANX2
E	LPCAT3	NOX4	PRNP	SLC3A2	TFR2	SAT1
F	MAP1LC3A	NQO1	PTGES2	SLC40A1	TP53	SQSTM1
G	MAP1LC3B	NRAS	RPL8	SLC7A11	TXNRD1	USP7
H	ACTB	GAPDH	HPRT1	18s	CD44	FSP1

**Table 2 cancers-15-00423-t002:** Primer sequence.

	Forward Sequence	Reverse Sequence
CD44	CCAGGCAACTCCTAGTAGTACAACG	CGAATGGGAGTCTTCTTTGGGT
SLC7A11	TTGGAGCCCTGTCCTATGC	CGAGCAGTTCCACCCAGAC
GAPDH	GCTCAGACACCATGGGGAAG	TGTAGTTGAGGTCAATGAAGGGG

## Data Availability

Data is contained within the article or Appendix A.

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
