# Peer review of "Sodium Butyrate Induces CRC Cell Ferroptosis via the CD44/SLC7A11 Pathway and Exhibits a Synergistic Therapeutic Effect with Erastin"

_cancers, 2023, doi:10.3390/cancers15020423_

Round 1
Reviewer 1 Report
In this manuscript, the authors examine the mechanisms underlying the anti-tumour effect of sodium butyrate (NaB) on colorectal cancer (CRC). NaB has been previously shown to inhibit CRC and thus here the authors aim at gaining mechanistic insight into this with a focus on the role of ferroptosis. The authors have previously shown that NaB can induce autophagy and apoptosis in CRC cells.
NaB alone reduced tumor cell line HCT-116 viability– but did not really affect the non-tumor celline FHC. In combination with inhibitors of caspases, GPX4, RIPK1 and autophagy, NaB promoted cell death was inhibited the best by the caspase and ferrostatin inhibitors (z-VAD and Ferrostatin). In HCT-116 cells GSH content was reduced by NaB and this was prevented by the ferroptosis inhibitors. These effects were not observed in the FHC cells, suggesting a tumor-specific signaling.
in HCT116 cells PCR array analysis showed that NaB induced transcriptional changes of genes related to iron and GSH metabolism. The authors focused on 2 genes: SLC7A11 and CD44 due to previous reports suggesting their role in the import of extracellular cysteines that are needed for the synthesis of GSH. In HCT116 cells NaB reduced the expression of both genes.
Analysis of patient transcription data suggests that increased expression of both CD44 and SLC77A11 occurs in CRC patient samples irrespective of disease stage. Increased expression of both genes also correlated with shorter survival times.
By western blotting, the authors show that in HCT-116 cells both CD44 and SLC7A11 expression levels are reduced in response to NaB. The authors then over-express CD44 in HCT116 cells and show that even with the CMV promoter the mRNA of both CD44 and SLC7A11 are downregulated by NaB. At the protein level upregulation of CD44 induced the upregulation of SLC7A11. Overexpression of CD44 also protected the cells from NaB induced cell death and increased GSH levels. Finally, the authors show that ferroptosis induction by Erastin was enhanced by increasing concentrations of NaB, only in HCT116 cells and not in FHC cells. Erastin promoted the reduction in GSH in HCT116 cells that correlated with a reduction in SLC7A11 as well.
Based on these experiments the authors conclude that in addition to inducing apoptosis in CRC tumor cells NaB can also promote ferroptosis. These data are based on only using these 2 cell lines and transcriptional analysis of primary human CRC patient data.
In general, I agree with the conclusions of the authors, but there are some details in the experiments shown that need to some additional work. And if possible, it would be nice to show the mechanism using other cell-lines in order to conclude that the phenotypes observed are not cell-line specific.
Specific comments:
Figure 1: Nec1 is not a specific inhibitor for RIPK1 induced necroptosis. The authors should please use Nec1s since Nec1 also inhibits IDO (doi: 10.1038/cddis.2012.176).
Figure 3: In (A) Please add the gene names analyzed into the figure panel. The gene names are also not in the figure legend so it is not clear which gene is which. Also, it is not clear what the red and grey bars are. In (C) please also add figure labels to the survival plots.
Figure 4:
Could the authors use some other gene than GAPDH as a control? GAPDH expression levels can also be subject to metabolic changes. Perhaps cytoskeletal or ribosomal genes? The confocal imaging analysis is not convincing. Both SLC7A11 and CD44 appear in general cytosolic and the image resolution provided does not allow for discerning co-localization or not. Also, the data provided for the in vivo analysis of the AOM/DSS model is really insufficient to reach any conclusions about this experiment. No information is provided on the NaB dosing, the number of tumors formed and the timing of the whole experiment, etc.
Figure 5:
In A, is this mRNA expression level relative to a house keeping gene?
In B, which band is specific for CD44?
Surprising that NaB would be able to override the strong CMV promoter of the pCDNA3.1 vector to reduce transcription of CD44. Is this to be expected?
Figure 6:
B: all images appear blurry
Author Response
第1点:图1:Nec1不是RIPK1诱导的坏死性凋亡的特异性抑制剂。作者应该使用Nec1s,因为Nec1也抑制IDO(doi:10.1038 / cddis.2012.176)。
回应1:感谢您的专业评论。我们仔细阅读了您推荐的文献,正如您提到的,Nec-1s 在 RIPK1 抑制方面相当于 Nec-1。然而,Nec-1s是一种更具特异性的RIPK1抑制剂,因为它缺乏IDO靶向。这些知识非常有助于我们更深入地了解体外或体内坏死抑制剂的使用。我们还检索了目前与IDO和铁死亡相关的研究,发现只有极少数研究表明IDO可能对组织或细胞的铁死亡有影响(目前关于结直肠肿瘤的研究数量为0)。此外,许多现有研究已经调查了Nec-1作为坏死抑制剂在铁死亡情况下的使用(例如,DOI:10.1073 / pnas.1415518111)。基于上述背景,我们认为使用Nec-1作为坏死抑制剂在铁死亡研究中仍然是可行的。
要点2:图3:在(A)中,请将分析的基因名称添加到图面板中。基因名称也不在图例中,因此不清楚哪个基因是哪个。此外,目前尚不清楚红色和灰色条是什么。在(C)中,请在生存图中添加图形标签。
回应2:非常感谢您的评论。我们替换了稿件中相应的图片,使图片和标签更易于理解。
要点3: 图4:作者可以使用GAPDH以外的其他基因作为对照吗?GAPDH表达水平也可能受到代谢变化的影响。也许是细胞骨架或核糖体基因?共聚焦成像分析并不令人信服。SLC7A11和CD44都出现在一般的胞质中,所提供的图像分辨率不允许辨别共定位与否。此外,为AOM / DSS模型的体内分析提供的数据确实不足以得出有关该实验的任何结论。没有提供关于NaB剂量,形成的肿瘤数量和整个实验的时间等信息。
回应3:感谢您的专业评论。为了便于理解,下面列出了对这一点的主要回应:
#1:我们最初的想法是使用GAPDH作为蛋白质测定的内部参考,以便与PCR测定保持一致。然而,正如你提到的,GAPDH表达水平也可能受到代谢变化的影响。然而,考虑到尽管有这个因素,GAPDH仍然是一种稳定可靠的细胞内参考,并已用于涉及HCT-116细胞的铁死亡研究(DOI:10.1002 / advs.202203357)。我们认为,GAPDH作为本研究的内部参数仍然是可信的。
#2:抱歉,我们可能无法提供更清晰的共聚焦图像,因为我们已经检查了我们拍摄的所有图像,但结果是相似的。SLC7A11 / xCT兔多克隆抗体(proteintech #26864-1-AP)和CD44小鼠单克隆抗体(CST#3570)用于孵育。使用共聚焦显微镜(奥林巴斯FV3000)拍摄照片。
#3:我们小组建立了AOM/DSS诱导的鼠炎性CRC模型。结果表明,NaB降低了AOM / DSS诱导小鼠的肿瘤负荷和肿瘤大小。将NAB剂量设置为添加到小鼠饮用水中的0.1M NaB。这些结果已在线发布(https://doi.org/10.1016/j.jff.2021.104862)。为了避免重复使用结果,本研究没有再次呈现这些数据,而是继续检查与铁死亡相关的指标。文中增加了对这部分文献的引用和解释。修订后的部分在论文中以蓝色标记。
LINE343-346:“为了验证NaB在体内的作用,我们建立了AOM / DSS诱导的小鼠炎症CRC模型。NaB可降低AOM / DSS诱导小鼠的肿瘤负荷和肿瘤大小。相应的结果和实验方法在我们之前的研究中得到了体现。
要点4: 图5:
在A中,这个mRNA表达水平是相对于管家基因的吗?
在 B 中,哪个条带特定于 CD44?
令人惊讶的是,NaB能够覆盖pCDNA3.1载体的强CMV启动子以减少CD44的转录。这是意料之中的吗?
回应4:感谢您的专业评论。为了便于理解,下面列出了对这一点的主要回应:
#1:是的,在本文中的所有RT-PCR结果中,靶基因的表达相对于管家基因(GAPDH)的表达。
#2:由于在这些结果中使用了CD44多克隆抗体,因此出现了多个条带。根据抗体规格和与抗体制造商的确认,分子量为80-95 kDa的条带是正确的条带。我们已经更正了结果部分中的条带。
#3:非常感谢您的评论。我们的结果表明,NaB对HCT-116细胞中的CD44 mRNA水平具有很强的调节(当5mM NaB处理24h时,CD44 mRNA的表达约为原始水平的6%-10%)。目前,尚无其他CD44在HCT-116细胞中过表达和同时NaB处理的结果可以给我们参考。本研究基于pCDNA3.1载体的过表达部分逆转了NaB的作用,NaB的作用不仅可能受到NaB的作用影响,还受到转染成功率等因素的影响。
Point 5: Figure 6:
B: all images appear blurry
Response 5: Thanks for your professional comments. Since we used a lower resolution image when uploading the first draft due to file size limitations, we have now replaced the image.
Reviewer 2 Report
In this study, the author demonstrated that the treatment of Sodium butyrate (NaB) induces ferroptosis in CRC cells through the CD44/SLC7A11 signaling pathway, and has synergistic effects with Erastin. Good writing and thinking logically. However, there are some major weaknesses of this study that would have to be addressed before this acceptable for publication.
Comments:
1. To more precisely demonstrate that NaB triggers ferroptosis in CRC cells, the other CRC cancer cell line should be used in this study.
2. What kinds of experiments was performed in figure 5E? We could not see any information in section 3.5 or figure legend of fig 5. This part should be corrected.
3. The resolution of Figure 2C is too low, the author should provide high-resolution picture to see the detail results of GO analysis.
4. In figure 4D, the quantification of CD44 and SLC7A11 expression should be addressed.
5.Following the minor Q2, what kinds of NaB delivery was used in vivo? There were two different NaB delivery methods in authors’ previous study. This part should be addressed in figure legend or material section.
Author Response
Point 1: To more precisely demonstrate that NaB triggers ferroptosis in CRC cells, the other CRC cancer cell line should be used in this study.
Response 1: Thanks for your professional comments. Your suggestion is very helpful for the rigor of our later experimental design, but at present, due to the time limit for the revision of the article, we are unable to complete the required experimental supplement. In fact, we have selected SW620 cells for verification, but the results show that SW620 is significantly less sensitive to NaB treatment than HCT-116, so we did not retain this part of the results. At the same time, as a typical KRAS mutant tumor cell, HCT-116 plays a more important role in the study of drug resistance and metastasis of colorectal cancer (DOI:10.7150/THNO.44705). Therefore, although only one tumor cell line was selected in this study, it is still persuasive to a certain extent.
Point 2: What kinds of experiments was performed in figure 5E? We could not see any information in section 3.5 or figure legend of fig 5. This part should be corrected.
Response 2: Thanks for your professional comments. We have corrected the description of figure 5E in section 3.5 and figure legend of fig 5. Revised portions are marked in blue in the paper.
LINE364-366: “The results showed that CD44 overexpression significantly restored the proliferation activity and cell morphology of cells treated with NaB for 24 h (Figure 5C, D, E).”
LINE606-608: “(D, E) Microscopy showing cell morphology and cell death. D: phase-contrast; E: (PI) staining. “
Point 3: The resolution of Figure 2C is too low, the author should provide high-resolution picture to see the detail results of GO analysis.
Response 3: Thanks for your professional comments. Since we used a lower resolution image when uploading the first draft due to file size limitations, we have now replaced the image.
Point 4: In figure 4D, the quantification of CD44 and SLC7A11 expression should be addressed.
Response 4: Thanks for your professional comments. We found that NaB could reduce the IHC staining of CD44 and SLC7A11 in tumor sites of mice. However, it is a pity that we have not had time to quantitatively analyze our IHC results. This is mainly due to the change in work plans caused by the recent popularity of COVID-19 in Guangzhou and Shanghai.
Point 5: Following the minor Q2, what kinds of NaB delivery was used in vivo? There were two different NaB delivery methods in authors’ previous study. This part should be addressed in figure legend or material section.
Response 5: Thank you very much for your professional comments and your attention to our previous work. Our previous work shows that NaB reduces tumor load and tumor size in AOM/DSS-induced mice. The NAB dose was set at 0.1 M NaB (o.p.) and 1g/kg NaB(i.p.)respectively. These results have been published online (https://doi.org/10.1016/j.jff.2021.104862). In order to avoid repeated use of the results, the study did not provide this data again, but selected mouse samples taken orally with NaB to further test ferroptosis-related indicators. Citations and explanations of this section of literature have been added to the text. Revised portions are marked in blue in the paper.
LINE343-346: “To verify the role of NaB in vivo, we established an AOM/DSS-induced murine inflammatory CRC model. 0.1M NaB(oral administration) reduces tumor load and tumor size in AOM/DSS-induced mice. Corresponding results and experimental methods are shown in our previous study."
Round 2
Reviewer 2 Report
Authors provided the clear response to my question. I think that this revised paper is acceptable for the publication.